# CEDAR: A Counter-Example Driven Agent with Regular Restriction

## Abstract

We introduce CEDAR, a Counter-Example Driven Agent with Regular Restriction in Minecraft, which learns and encodes informal specifications and skills as regular languages. Large language models (LLMs) interpret intent and help select a compact, task-specific sub-alphabet; deterministic finite automata (DFAs) provide a canonical, composable temporal scaffold for execution and checking. CEDAR learns skill DFAs via active grammatical inference, reuses them by *template adaptation* (verb-fixed, noun-substituted alphabets), and enforces human constraints by DFA composition (e.g., intersection with "sleep at night"). A counterexample loop ties together environment logs, specifications, and learned DFAs under a probabilistic membership oracle. Across Minecraft and iTHOR, CEDAR improves controllability and amortized efficiency over program-generating agents under matched wall-clock and LLM-query budgets, while enabling symbolic skill retrieval and verification.

## 1 Introduction

LLM-based agents have achieved significant success in control and planning within complex open-world environments Wang et al. (2023a); Liu et al. (2024b); Zhu et al. (2023); Liu et al. (2023); Yan et al. (2023); Yao et al. (2023b); Tsai et al. (2023); Wang et al. (2024b). Early research explored using LLM-generated structured programming techniques to enhance robotic manipulation and gameplay Liang et al. (2023); Sun et al. (2020); Zhao et al. (2021); Singh et al. (2022); Wang et al. (2024a). To improve the quality of the generated code, researchers are incorporating environment feedback Huang et al. (2022); Shinn et al. (2023), advanced prompts Wei et al. (2023); Yao et al. (2023b), and external knowledge retrieval Wang et al. (2024c); Zhu et al. (2023).

Despite these advancements in control, planning remains a significant challenge in open-world environments Kolve et al. (2017a); Fan et al. (2022); Puig et al. (2023). Various planning approaches have been developed, such as task decomposition Wang et al. (2023a); Zhu et al. (2023), elaborate prompts Wang et al. (2024b); Zhang et al. (2023), multi-modal information Qin et al. (2024); Zheng et al. (2023); Wang et al. (2023b); Zhao et al. (2024), and skill management Wang et al. (2023a); Yuan et al. (2023); Zhu et al. (2023). Goal completion is a common way to evaluate the effectiveness of these planning methods in open-world environments Wang et al. (2023a); Zhu et al. (2023); Wang et al. (2023b), as it requires understanding natural language and mapping high-level commands to precise, executable actions in specific contexts. However, there is currently no way to logically verify if the LLM-generated executable policy fully understands and obeys human specifications, potentially leading to unexpected or harmful results Yao et al. (2024); Gu et al. (2024); Moos et al. (2022).

To ensure that LLM-generated executable policies adhere to human instructions and bridge the gap between natural and regular language, we implement a logic verifier. This is complemented by methods like autoformalization Wu et al. (2022); Giannakopoulou et al. (2021) and LLM-based automata learning Vazquez-Chanlatte et al. (2024); Alsadat et al. (2024); Chen et al. (2024). In this paper, we utilize LLM-based automata learning to formalize informal specifications and address the challenge of planning in open-world environments while adhering to human specifications.

To achieve this, we introduce CEDAR, a Counter-Example Driven Agent in Minecraft that learns skills through DFA learning to align with informal specifications. CEDAR consists of three main components: 1. **DFA Learner**, which learns skills in the form of DFAs based on formalized human

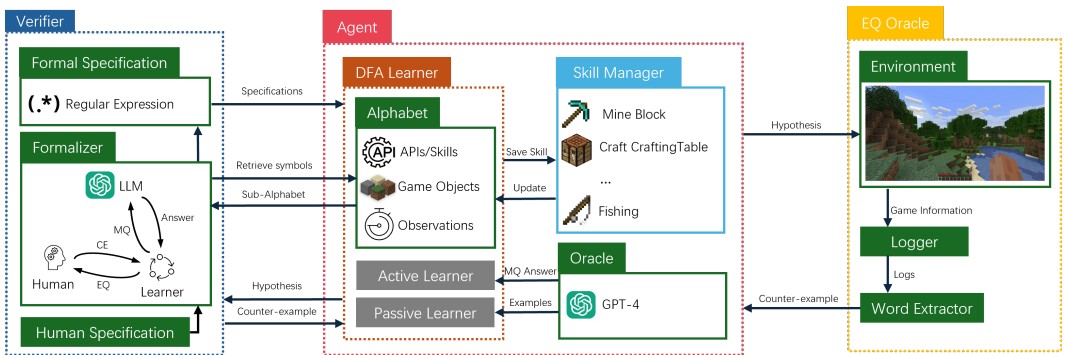

Figure 1: **CEDAR Workflow**. CEDAR is built around three essential components: 1. DFA Learner which leverages active learning algorithms for continuous, lifelong learning and goal-directed skill acquisition, to construct DFAs representing various skills. 2. Skill Manager which manages the repository of learned skills and adapts them to new tasks by adjusting the DFA's alphabet as needed. 3. Verifier which ensures that the DFAs learned by the system conform to human specifications. It converts these natural language specifications into DFAs and then cross-checks them against the skill DFAs to detect any discrepancies.

specifications utilizing DFA learning algorithms. The LLM oracle answers membership queries for active DFA learners. 2. **Skill Manager** that is responsible for storing the learned skills and extending them to new tasks by modifying the alphabet of the corresponding DFA. 3. **Verifier**, which takes human specifications as input and formalizes them into DFAs using a human-in-the-loop DFA learning paradigm where the human provides counterexamples to the hypothesis learned by an LLM. The output formal specifications are given to the DFA learner. Once the DFA learner has learned a skill, the verifier checks if this hypothesis DFA violates any formalized human specifications and provides a counterexample.

We encode both skills and human specifications as DFAs over a compact, LLM-assisted sub-alphabet; the environment supplies counterexamples while the LLM answers (noisy) membership queries, and we refine policies until execution and specifications agree. DFAs give CEDAR three concrete advantages over program-generation baselines such as VOYAGER Wang et al. (2023a): (i) a minimal canonical structure with closure under intersection for constraint enforcement; (ii) an explicit failure interface (counterexamples) that repairs skills; and (iii) *template adaptation*—reusing verb-specific automata by substituting object/event symbols—enabling data-efficient transfer.

**Our main contributions are:** (i) A practical *regular-language scaffold* for embodied skills: LLMs ground intent; DFAs supply verifiable temporal structure. (ii) *RAG-based sub-alphabet construction* that makes learning tractable in large action/object/event spaces. (iii) *Template-DFA adaptation* and a *bidirectional counterexample loop* that keep specifications, skills, and environment behavior consistent under noisy oracles.

## 2 RELATED WORK

**Active DFA learning.** Classical active automata learning (e.g., L* Angluin (1987)) identifies a target DFA using *membership queries* (MQs) and *equivalence queries* (EQs) posed to a (minimally adequate) teacher. Numerous refinements study counterexample processing, efficiency, and practical robustness; passive state-merging methods such as RPNI/EDSM Lang et al. (1998); Oncina & García remain influential but require labeled corpora rather than interaction. Recent work treats noisy oracles and probabilistic teachers, introducing consistency mechanisms and query budgets; our instantiation follows the probabilistic MAT (pMAT) view with LAPR for cache consistency under stochastic MQs Chen et al. (2024). In CEDAR (Sec. 3.2), the environment implements the EQ oracle and the LLM implements the MQ oracle, enabling on-policy data collection and counterexample-driven refinement.

A growing line of work uses LLMs to induce or interact with *formal* structure. Closest to us are approaches that use LLMs as (noisy) teachers for regular languages—answering MQs, proposing candidate DFA structure, or summarizing traces into symbolic events Chen et al. (2024). Orthogonal but complementary are methods translating natural language into *temporal logics*, e.g., NL2LTL Fuggitti & Chakraborti (2023), data-efficient NL→LTL for robot tasking Pan et al. (2023), and Lang2LTL-2 for grounded spatiotemporal commands Liu et al. (2024a). These typically perform one-shot translation and do not execute or refine policies online; they also fixed to one certain area (e.g. where the pre-train dataset from); CEDAR instead *executes and repairs* DFAs via counterexamples, while remaining compatible with LTL front-ends (e.g., compiling LTL monitors to regular abstractions when possible).

ReAct Yao et al. (2023a) synergizes reasoning traces with tool-use to interleave "thought" and "act." It improves sample-efficiency and transparency in web and question-answering settings, but does not endow agents with canonical temporal controllers or formal operations over behaviors. Voyager Wang et al. (2023a) pioneered open-ended skill discovery in Minecraft by prompting an LLM to synthesize executable JavaScript skills, storing them in a growing library with automatic curriculum and retrieval. While powerful, code-based skills lack canonical structure, making formal verification and composition (e.g., intersection for constraint enforcement) difficult. CEDAR replaces free-form programs with minimal DFAs over a task-specific alphabet obtained via RAG (Fig. 2), enabling *symbolic* retrieval, exact composition (intersection/concatenation), and online repair through counterexamples (Sec. 3.4).

ADAM Yu & Lu (2025) advocates causal graphs as intermediate structure for embodied agents in open worlds. Such structure is complementary to our RAG-based alphabet construction: causal relations can guide symbol discovery (e.g., prerequisite events) and suggest safer exploration policies before DFA synthesis. SELP Wu et al. (2025) demonstrates majority-group voting schemes to validate LLM-generated plans. In our setting, similar aggregation can be used to denoise MQ answers before caching (cf. LAPR), or to vet counterexample explanations prior to updating the hypothesis DFA.

Compared to code-generating agents (e.g., Voyager), CEDAR trades some expressivity for *canonicality, verifiability, and compositionality*. Compared to NL→LTL translators, CEDAR targets *online* policy learning and repair with environment-supplied counterexamples, while remaining compatible with temporal-logic front-ends. Relative to ReAct-style planners, CEDAR supplies an explicit, minimal controller with closure properties and a clear failure semantics (rejecting states and product constructions). Finally, while classic passive learners (RPNI/EDSM) inform our background, CEDAR's operational loop is fully *active*, leveraging a probabilistic teacher and an environment EQ oracle to support both goal completion and continual (lifelong) acquisition.

## 3 METHOD

### 3.1 LLM–ASSISTED DFA LEARNING

**Notation.** Let $\mathcal{AP}_{\text{act}}, \mathcal{AP}_{\text{obj}}, \mathcal{AP}_{\text{evt}}$ be the global sets of action, object, and event–monitor symbols. The global alphabet is $\Sigma_{\text{global}} := \mathcal{AP}_{\text{act}} \uplus \mathcal{AP}_{\text{obj}} \uplus \mathcal{AP}_{\text{evt}}$ (disjoint union). For a task, Retrieval–Augmented Generation (RAG) returns a finite sub-alphabet $\Sigma \subseteq \Sigma_{\text{global}}$. A *word* is a sequence $w \in \Sigma^*$ extracted from logs; a *language* is $L \subseteq \Sigma^*$. We denote the stochastic membership oracle by $\mathcal{O}_{\text{MQ}} : \Sigma^* \to \{0, 1\}$ and maintain caches $C_{\text{MQ}}, C_{\text{EQ}} \subseteq \Sigma^* \times \{0, 1\}$ for answered MQs and EQ labels/counterexamples.

LLMs map informal instructions to semantic neighborhoods (verbs, objects, conditions) but do not provide canonical temporal structure or guarantees. DFAs provide minimal, canonical automata with closure properties and efficient checking. CEDAR fuses the two: LLMs select a compact sub-alphabet and answer noisy membership queries; DFAs scaffold execution and verification; counterexamples circulate between the environment and specifications to repair both.

**Sub-alphabet construction (RAG).** Because the Minecraft interface exposes many APIs, skills, and $> 1000$ objects, selecting $\Sigma$ is a specification-decomposition step. Each symbol in $\Sigma_{\text{global}}$ has a textual description. Given a human specification, we embed it and retrieve symbol candidates by cosine similarity; top candidates from $\mathcal{AP}_{\text{act}}, \mathcal{AP}_{\text{obj}}, \mathcal{AP}_{\text{evt}}$ are injected into prompts so the LLM

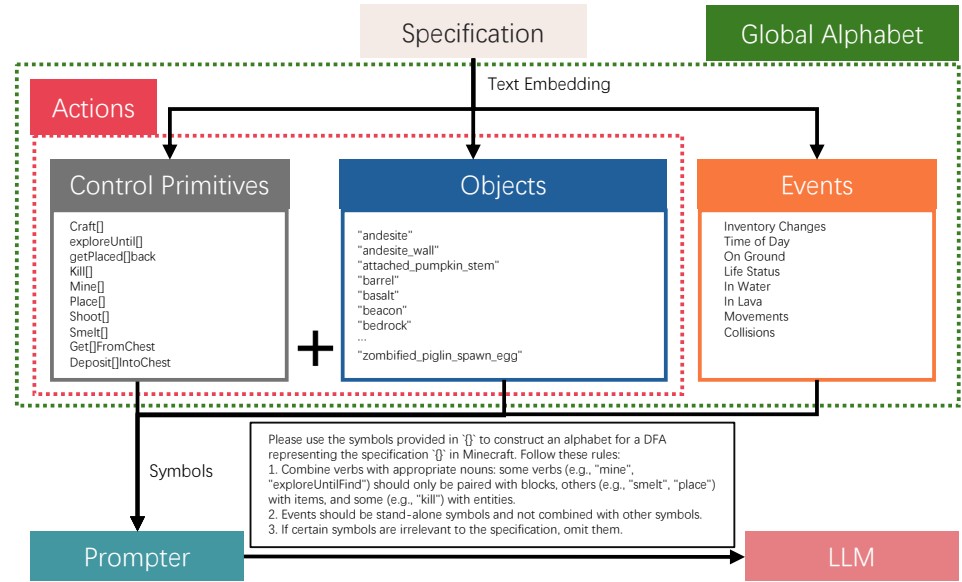

Figure 2: **Global alphabet and RAG**. From $\Sigma_{\mathrm{global}}$ (control primitives, objects, events), RAG retrieves task-relevant symbols to form a sub-alphabet $\Sigma$.

proposes $\Sigma$ (details in Appx. A.4). If $\Sigma$ is incomplete, the target DFA cannot be learned (often yielding no accepting state in practice). We analyze the erroneous DFA and reuse EQ counterexamples to refine retrieval and expand $\Sigma$ (Fig. 2).

**Finite abstractions.** Although DFAs cannot count unboundedly, many tasks admit *milestone* monitors (e.g., has_$\geq$3_cobblestone) that render a finite $\Sigma$. We encode bounded accumulation via such monitors.

## 3.2 DFA LEARNER

We learn DFAs *actively* via queries in a probabilistic MAT (pMAT) setting Angluin (1987); Chen et al. (2024). An active learner maintains a DFA hypothesis and interacts with two oracles: (i) an LLM that answers membership queries (MQs) stochastically, and (ii) an environment-backed equivalence oracle (EQ) that returns counterexamples when the hypothesis is wrong. We use LAPR Chen et al. (2024) to ensure cache consistency under noisy MQs.

We distinguish DFAs for *specifications* and for *skills*; the former are used by the verifier (Sec. 3.4), while this section focuses on skill learning with a single unified active loop.

**Active learning loop.** Given $\Sigma$, the learner repeatedly:

1. issues MQs on words $w \in \Sigma^*$; the LLM returns labels recorded in $C_{\mathrm{MQ}}$,

2. synthesizes/updates a DFA hypothesis consistent with $C_{\mathrm{MQ}}$ under LAPR,

3. queries the EQ oracle by executing the hypothesis in a wrapper environment that logs events and a word extractor that maps logs to words; any mismatch yields a labeled counterexample added to $C_{\mathrm{EQ}}$,

4. if counterexamples indicate missing symbols, re-query RAG to refine $\Sigma$ and restart the loop on the expanded alphabet.

**Goal completion.** The loop terminates when the environment confirms the goal has been achieved and no counterexamples are found within the time/interaction budget. The resulting DFA is stored in the Skill Manager.

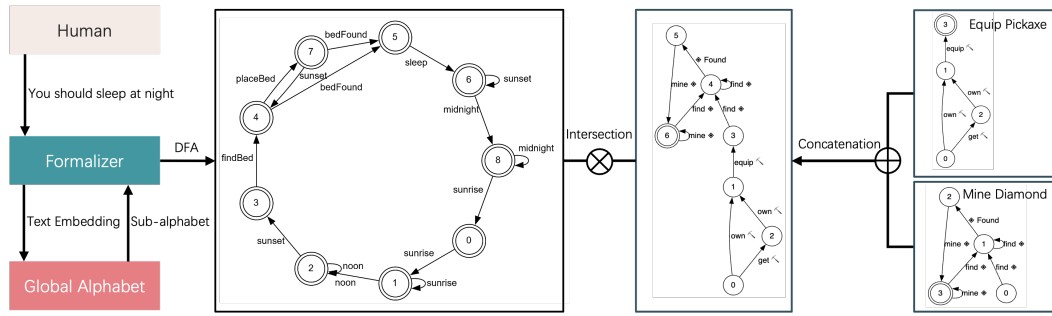

Figure 3: DFA Intersection Operation: The intersection creates a new DFA that accepts only the words accepted by both original DFAs. The top DFA represents the specification "Please sleep at night," while the bottom DFA corresponds to the skill "Mine `diamond_ore`." Edges to rejecting states are omitted for clarity.

**Lifelong learning.** CEDAR continues *beyond* a single goal: after completion, it discovers the next goal, constructs a new sub-alphabet via RAG, and re-runs the same active loop to produce a new DFA. We initialize action symbols with control primitives aligned with VOYAGER Wang et al. (2023a) (lightly renamed to improve prompting). Actions outside the current $\Sigma$ default to rejection until RAG expands $\Sigma$ in response to counterexamples. This continual process accumulates a library of DFAs that grow in scope and robustness.

### 3.3 SKILL MANAGER

A skill is stored as a tuple $\langle \mathcal{A}, v, n, E, D \rangle$, where $v \in \mathcal{AP}_{\text{act}}$ is a verb, $n \in \mathcal{AP}_{\text{obj}}$ is an object, $E \subseteq \mathcal{AP}_{\text{evt}}$ is the set of success events, and $D \subseteq \Sigma^* \times \{0, 1\}$ collects labeled evidence used to construct $\mathcal{A}$. For learned DFAs,

$$D = \{(w, y) \mid (w, y) \in C_{\text{MQ}}\} \cup \{(w, y) \mid (w, y) \in C_{\text{EQ}}\}.$$

We write the induced positive and negative sets as $D^+ = \{w \mid (w, 1) \in D\}$ and $D^- = \{w \mid (w, 0) \in D\}$. To use DFAs as policies, we execute actions along the shortest path from the current state to an accepting state. If an invoked action is absent from logs, we treat it as a failure, temporarily remove the corresponding edge, and recompute a shortest accepting path.

**Retrieval and templating.** Given a query $(v', n')$, the manager matches stored $(v, n)$ skills. If both match, we return the skill. If $v' \neq v$, no skill is returned. If $v' = v$ but $n' \neq n$, we retrieve all skills with verb $v$ and pass them as context to the LLM to select a template DFA; the noun-specific sub-alphabet is substituted and all transition symbols $\delta^{\mathcal{A}}$ (and examples in $D$) are updated accordingly. The modified DFA is then refined by the active learner rather than trained from scratch.

### 3.4 VERIFIER

The verifier ensures that learned skills align with human specifications (goals and constraints given in natural language). It translates each specification into a regular language via *active* DFA learning: the LLM answers MQs, while humans serve as the EQ oracle to validate counterexamples and maintain intent; LAPR keeps MQ/EQ caches consistent. Alphabet selection for each specification uses the same RAG pipeline as skills (see Fig. 1 and Appx. A.4).

There are two main advantages to representing human specifications and skills as DFAs. First, DFAs derived from specifications can check compliance by matching words extracted from new logs. When a violation is detected, we merge alphabets and take the *intersection* of the skill and specification DFAs to obtain a compliant policy (Fig. 3). Formally, for skills

$$s_1 = \langle \mathcal{A}_1, v_1, n_1, E_1, D_1 \rangle, \quad s_2 = \langle \mathcal{A}_2, v_2, n_2, E_2, D_2 \rangle,$$

their conjunctive merge is

$$s_{\text{conj}} = \langle \mathcal{A}_{\cap}, v, n, E_1 \cup E_2, D_1 \cup D_2 \rangle,$$

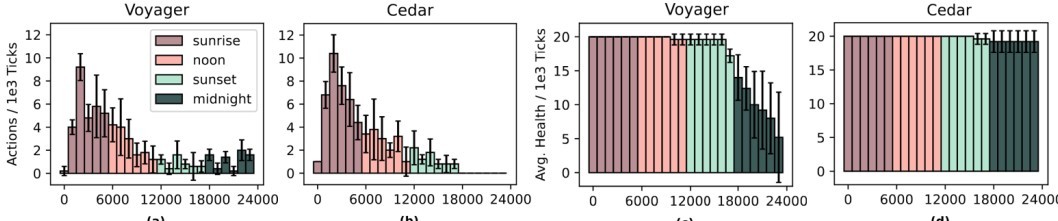

Figure 4: Comparison of action counts and average health across time for VOYAGER and CEDAR. The human instruction here is to "craft a diamond pickaxe and keep collecting diamonds. Please sleep at night. You are given a bed." (a) and (b) depict the number of actions per 1000 ticks for the VOYAGER and CEDAR agents; (c) and (d) show the average health of the agent per 1000 ticks for VOYAGER and CEDAR. The results were averaged over five trials that last three days each time on the same map.

with acceptance $w \in \mathcal{L}(\mathcal{A}_\cap) \iff w \in \mathcal{L}(\mathcal{A}_1) \wedge w \in \mathcal{L}(\mathcal{A}_2)$ and $\mathcal{A}_\cap = \mathcal{A}_1 \cap \mathcal{A}_2$ (product construction, accepting states $F_1 \cap F_2$).

Second, DFAs support *skill chaining* via concatenation, even without specifications. Let $\mathcal{A}_1$ (e.g., craft and equip a pickaxe) and $\mathcal{A}_2$ (mine a diamond). Concatenate by merging each accepting state of $\mathcal{A}_1$ with the initial state of $\mathcal{A}_2$ (state relabeling) to obtain

$$s_{\text{con}} = \langle \mathcal{A}_\circ, v, n, E_1 \cup E_2, D_1 \circ D_2 \rangle,$$

with $w \in \mathcal{L}(\mathcal{A}_\circ) \iff \exists x, y \in \Sigma^* : w = x \cdot y, \ x \in \mathcal{L}(\mathcal{A}_1), \ y \in \mathcal{L}(\mathcal{A}_2)$, and $D_1 \circ D_2 = \{ w_1 \cdot w_2 \mid w_1 \in \mathcal{L}(\mathcal{A}_1), w_2 \in \mathcal{L}(\mathcal{A}_2) \}$.

## 4 EMPIRICAL RESULT

In this section, we evaluate our method within the Minecraft game environment and the iTHOR simulator (Kolve et al., 2017b), demonstrating its advantages over the popular VOYAGER Wang et al. (2023a). We begin by assessing the CEDAR agent's ability to follow human instructions across various settings. Following this, we measure our method's performance in terms of the success rate in completing specific tasks. We then compare the lifelong learning efficiency of our method against VOYAGER. Finally, we test the generality of our approach by extending the learned skills to unseen tasks. The LLMs we used in the evaluation are `gpt-4o` for task decomposition and answering membership queries, `gpt-4o-mini` for JSON translation, and `text-embedding-3-large` for computing text embeddings. To ensure a fair comparison with VOYAGER, *human-provided counterexamples (CEs) are not used* in any task-completion experiment, including those reported in Tables 2 and 1. All CEs during evaluation are collected automatically from environment interaction via the EQ-oracle. Human-provided CEs are used only to refine high-level *constraint* DFAs in our specification-following demonstrations and are not required for learning task-specific skills.

### 4.1 HUMAN SPECIFICATION FOLLOWING STUDY

In the experiments focused on following human specifications, both the VOYAGER and CEDAR agents were given a goal with a specification to constrain the agent's policy. In real-world scenarios, agents often face potential dangers, represented here by randomly generated zombies at night in Minecraft. Using sleep to bypass the night is an effective strategy in such situations. For this experiment, the goal was to collect diamonds with the specification to sleep at night. The difficulty of the game is set to normal for monster generation. Both VOYAGER and CEDAR were spawned in the same location and world, and each was provided with a bed to eliminate the variable of bed crafting, allowing us to focus on how well each agent understands and follows the human specification. The results in Figure 4 demonstrate that CEDAR, which enforces strict adherence to human instructions using DFAs, successfully prevents the agent from working during midnight. Notably, the CEDAR agent maintains higher health levels during the night, reflecting its compliance with the sleep instruction, while VOYAGER chooses to contend with monsters spawned at night.

Table 1: Statistics on the action count and objects gained for our approach and popular MineCraft agent VOYAGER. The results are presented as mean ± standard deviation (successful trials / total trials).

| Method | Action Counts | Underground | Overground | Items | Gained Objects |
|---|---|---|---|---|---|
| VOYAGER | $106 \pm 5$ | $152 \pm 47$ | $50 \pm 10$ | $27 \pm 7$ | $229 \pm 44$ |
| CEDAR (Ours) | $138 \pm 10$ | $195 \pm 31$ | $136 \pm 18$ | $58 \pm 6$ | $388 \pm 36$ |

In Minecraft, having a well-crafted plan that guides the agent on what to do and when to do it is crucial for efficient exploration, as some activities are highly time-sensitive like villager trading and honey collection. In this experiment, we assigned the agents the goal of exploring the world with the specific instruction to mine minerals only at night. Since mining can be done at any time and typically involves minimal monster encounters if not digging in natural caves or mines, the safer daytime hours can be better utilized for other tasks. Figure 5 illustrates that CEDAR adheres to this instruction, optimizing the use of daytime for item collection and reserving nighttime for mineral extraction. In contrast, VOYAGER fails to follow the instruction, leading to inefficient use of daytime. VOYAGER frequently moves between underground and overground places, wasting time and resulting in fewer actions and items collected. The objects obtained by VOYAGER are irregular, whereas CEDAR predominantly collects underground blocks at night. Moreover, Table 1 shows the total amount of objects collected by CEDAR exceeds that of VOYAGER. These results demonstrate the effectiveness of CEDAR in better utilizing daytime opportunities by strictly following human instructions.

The spatial distribution of objects in Minecraft is highly dependent on biomes; staying within a specific biome can significantly enhance the collection speed of resources native to that biome. In this experiment, we instructed the agents to explore within a biome called windswept_forest. By integrating biome symbols into the sub-alphabet for learning human specifications and skills, CEDAR is able to comprehend biome information within game events and use it to constrain its activity area.

As shown in the agent activity area heatmap in Figure 6, the VOYAGER agent ignored the human specification of staying within the windswept_forest biome (the area in green) and traversed across different biomes. In contrast, the CEDAR agent effectively restricted its activities to the designated biome, adhering to the given instruction.

Both the VOYAGER and CEDAR agents had sufficient information observed from the Minecraft environment, yet VOYAGER failed to follow four types of human specifications. There are two main reasons for this failure. First, VOYAGER decomposes human specifications into sub-tasks rather than a set of constraints. This approach means that once the corresponding sub-task is completed, VOYAGER disregards it. In the first experiment shown in Figure 4, the VOYAGER agent did indeed sleep on the first night, but subsequently forgot this constraint and continued collecting dia-

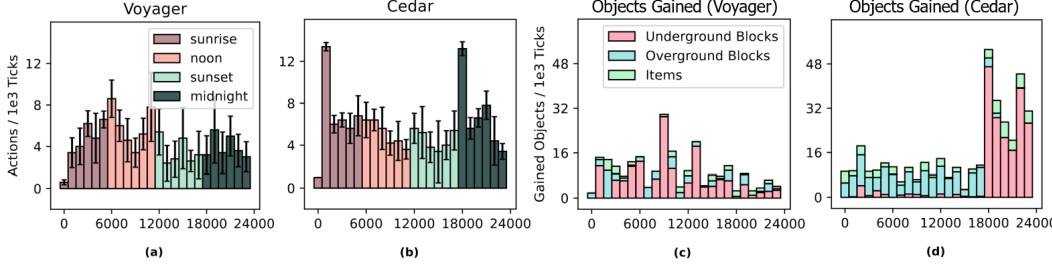

Figure 5: Comparison of action counts and collected objects across time for VOYAGER and CEDAR. Subplots (a) and (b) depict the total number of actions per 1000 ticks for VOYAGER and CEDAR, respectively. Subplots (c) and (d) present the distribution of underground blocks, overground blocks, and items collected per 1000 ticks. The given instruction was "explore the world and collect as many different items as possible, but you can only dig for minerals like iron and diamond at night." The experiment was repeated on the same map and spawn location 5 times, with each trial lasting 3 days.

Table 2: Performance comparison between VOYAGER and CEDAR across different crafting tasks. The results are presented as mean ± standard deviation (successful trials / total trials). The values represent the mean and standard error of the prompting iterations, and the fractions indicate the number of goal completions out of total trials. The tasks to the left of the second vertical line are included in the skill library (S.L.) for both agents. w/o S.L means it starts without skill library.

| Method | Wooden Pickaxe | Iron Pickaxe | Diamond Pickaxe | Lava Bucket | Compass |
|---|---|---|---|---|---|
| VOYAGER w/o S.L. | $7 \pm 2$ (5/5) | $29 \pm 6$ (5/5) | $35 \pm 12$ (2/5) | $29 \pm 9.6$ (4/5) | $26 \pm 2.9$ (3/5) |
| VOYAGER | $\mathbf{4.4 \pm 2.5}$ (**5/5**) | $17 \pm 3.5$ (5/5) | $26 \pm 11$ (3/5) | $23 \pm 5.4$ (5/5) | $18 \pm 1.5$ (5/5) |
| CEDAR w/o S.L. | $6 \pm 3$ (5/5) | $31 \pm 3$ (5/5) | $41 \pm 11$ (3/5) | $28 \pm 4.5$ (5/5) | $29 \pm 2.5$ (2/5) |
| **CEDAR (Ours)** | $6 \pm 3$ (5/5) | $\mathbf{11 \pm 5.5}$ (**5/5**) | $\mathbf{20 \pm 6.5}$ (**5/5**) | $\mathbf{10 \pm 7.7}$ (**5/5**) | $\mathbf{10 \pm 2.1}$ (**5/5**) |

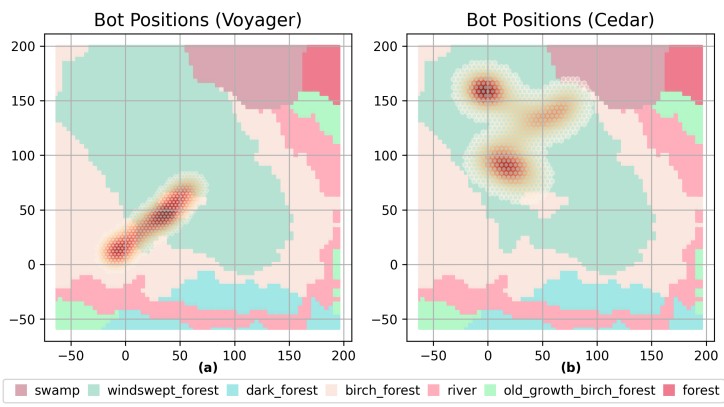

Figure 6: The background colors denote various biomes, and the heatmap overlay represents the bot's activity. CEDAR follows the human instruction to "explore the world but stay in the windswept forest." The heatmap intensity indicates the frequency of the bot's activities, with deeper colors representing areas of higher activity.

monds both day and night. In contrast, CEDAR learns the specification as a regular language, which continuously reinforces the instruction for the agent to sleep at night. Second, VOYAGER lacks a mechanism to ensure that the generated program fully adheres to human specifications. In contrast, CEDAR enforces that the DFAs of learned skills are free from counterexamples when tested against the DFAs of human specifications. This approach provides validation that the learned skills align with the given human specifications.

## 4.2 GOAL COMPLETION PERFORMANCE

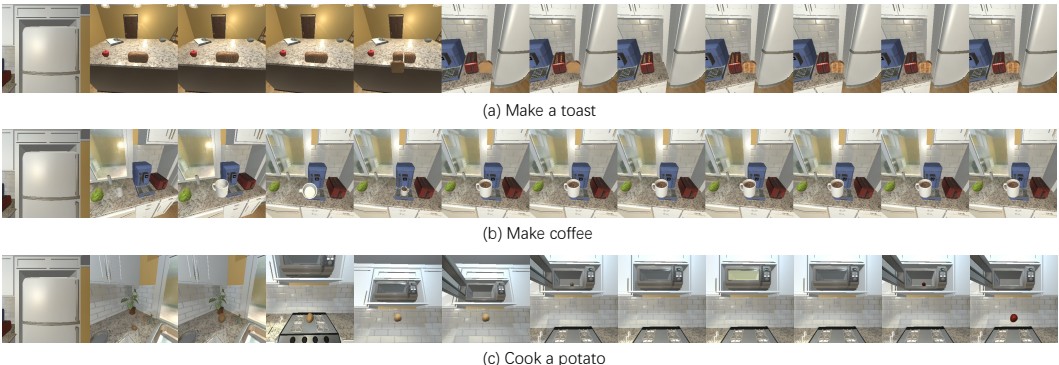

(a) Make a toast

(b) Make coffee

(c) Cook a potato

Figure 7: **iTHOR rollouts guided by DFAs.** Time-lapse frames (left → right) of three representative tasks executed in iTHOR using CEDAR: *top*—MAKE_1_TOAST (locate bread, place in toaster, toast); *middle*—MAKE_COFFEE (place mug, operate machine); *bottom*—COOK_POTATO (open microwave, place potato, heat). Rows are separated by black bars. Policies are induced from DFAs learned with an LLM MQ-oracle and an environment-backed EQ-oracle that supplies counterexamples.

**Minecraft Tasks.** We evaluated the goal-completion performance of our method by comparing success rates across different tasks with VOYAGER. The results presented in Table 2 underscore

two principal advantages of CEDAR: (1) the skills acquired by CEDAR exhibit greater robustness, and (2) CEDAR is capable of efficiently extending these learned skills to previously unseen tasks. CEDAR demonstrates efficiency in task resolution when the relevant skills are already included in the skill library, necessitating only a single LLM query to translate the goal into a regular language. For unseen tasks, CEDAR surpasses VOYAGER by extending the learned skills through straightforward modifications to the alphabet of the DFAs corresponding to those skills. However, a drawback of CEDAR is that it requires a greater number of LLM prompting iterations to accurately learn a DFA for a given skill. This is due to its iterative process of testing the DFA in the environment until no counterexamples remain, thereby requiring continuous querying of the LLM for additional examples.

**iTHOR Tasks**  To evaluate beyond Minecraft, we ran five iTHOR tasks—MAKE_1_TOAST, MAKE_COFFEE, COOK_POTATO, MAKE_SALAD, STORE_PLATE_IN_FRIDGE—using the standard CEDAR pipeline with RAG-based alphabet construction (no skill library, as tasks are simple). Figure 7 visualizes three representative rollouts: toast, coffee, and potato. Most tasks succeeded in one shot; the exception was COOK_POTATO, where unstable camera perspective caused a miss. The EQ-oracle surfaced a counterexample, after which inserting a LOOKUP action into the DFA enabled the agent to adjust its view and complete the task (bottom row of Fig. 7). LLM query costs for these experiments are reported in Appendix A.7.

## 5 CONCLUSION

This paper presents CEDAR, a Counter-Example Driven Agent with Regular Restrictions, developed for the Minecraft environment. CEDAR incorporates human specifications formalized as DFAs, enabling the agent to learn and refine skills in alignment with these specifications. By active DFA learning algorithms, the agent adapts to new tasks and improves existing skills through interaction with the environment. Empirical evaluations suggest that CEDAR offers improvements over prior methods such as VOYAGER, particularly in terms of controllability, robustness, and extensibility. The use of DFAs helps maintain adherence to human instructions, reducing the likelihood of unintended behaviors. Additionally, CEDAR's ability to extend learned skills to new tasks by modifying the DFA alphabet contributes to its adaptability in open-world settings. By integrating formal verification techniques with learning algorithms, this work explores how autonomous agents can be made more reliable and responsive to human-specified constraints in complex environments.

**Reproducibility Statement.**  We release an anonymous archive in the supplementary materials containing all code, configuration files, and scripts needed to reproduce our results, including the DFA learner (LAPR/pMAT instantiation), the word–extraction logger, the verifier, and the full Retrieval-Augmented Generation (RAG) implementation with exact prompts and embedding settings. We also provide cached MQ/EQ logs, precomputed DFAs used in figures/tables, and seeds/world identifiers for all runs to enable exact regeneration of reported numbers. Experimental protocols and evaluation metrics are specified in the main text (Sec. 3.2 and Empirical Result), with additional implementation details and RAG settings in Appendix A.4. All hyperparameters, random seeds, and query budgets are listed in the appendix and config files; scripts are included to regenerate every table and figure from raw logs. No human counterexamples are used in task-completion results (only environment-collected CEs), ensuring they can be reproduced end-to-end from the provided materials.

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

# A APPENDIX

## A.1 RUNTIME COMPARISON

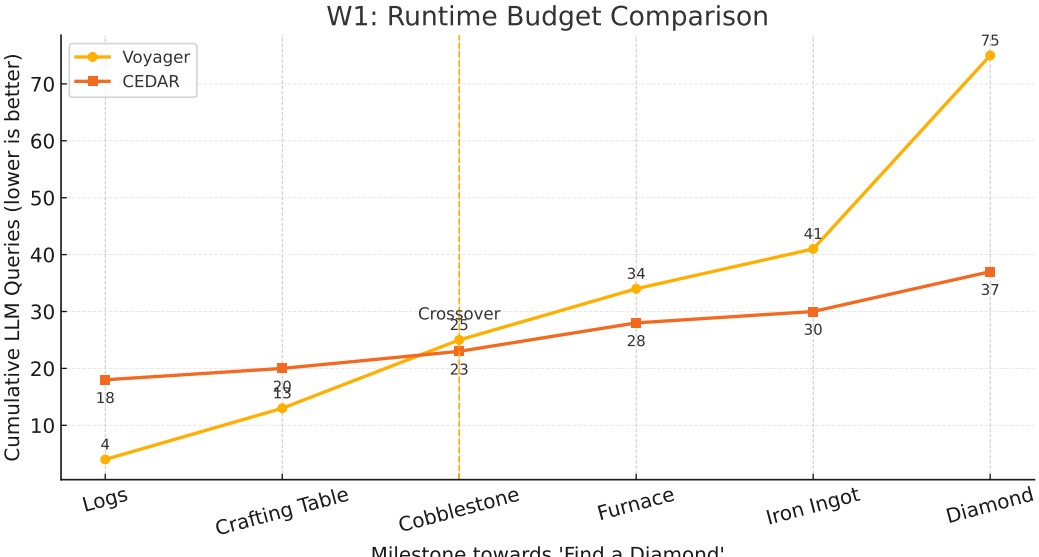

Figure 8: Runtime budget comparison (cumulative LLM queries) for the *Find a Diamond* task. Lower is better.

Figure 8 tracks cumulative LLM queries required to reach successive milestones on *Find a Diamond* (crafting table → cobblestone → furnace → iron ingot → diamond). Across the trajectory, CEDAR reaches each milestone with fewer accumulated queries than VOYAGER, and the gap widens at later stages. This reflects two design advantages of CEDAR: (i) the RAG–restricted sub-alphabet reduces query fan-out when proposing actions or verifying preconditions, and (ii) the counterexample-driven refinement prevents repeated prompt chains on already-disproved behaviors. Practically, this yields lower amortized query cost for long-horizon tasks while preserving reliability—CEDAR spends fewer queries "figuring out" what to do and instead reuses validated automata structure to progress efficiently.

**Alphabet construction efficiency.** To reduce runtime overhead during DFA construction, we adopt a modular retrieval strategy that independently selects *verbs* and *nouns*. Minecraft exposes roughly ten core control primitives (verbs), enabling lightweight LLM querying for verb choice. For nouns (game objects), we perform embedding-based retrieval and keep the top-20 candidates given the task description. These sets are combined to form a compact sub-alphabet, substantially shrinking the search space and minimizing LLM usage during DFA synthesis.

**Wall-clock comparison.** Table 3 reports mean $\pm$ standard deviation of stage-wise execution time (in seconds) for VOYAGER and CEDAR on our benchmark protocol. Although VOYAGER is slightly faster in several micro-stages, it typically requires multiple LLM refinement iterations, increasing its true interaction cost. In contrast, CEDAR constructs and verifies DFA-based skills once and then reuses them symbolically, avoiding repeated LLM calls. Skill retrieval is also faster and more robust due to symbolic verb–noun matching; the larger variance stems from occasional fallback prompts when noun mismatches occur (see Section 3.2).

| Stage | VOYAGER | CEDAR |
|---|---|---|
| Task Decomposition | 2.749 s ± 1.238 | 3.943 s ± 2.134 |
| Code/Sample Generation | 6.381 s ± 1.990 | 5.548 s ± 1.876 |
| Program Description | 2.384 s ± 1.208 | N/A |
| Skill Addition | 2.653 s ± 1.228 | 0.021 s ± 0.008 |
| DFA Construction | N/A | 18.548 s ± 14.289 |
| Skill Retrieval | 0.323 s ± 0.235 | 0.089 s ± 0.586 |
| **Total Execution Time** | **62.427 s** ± 55.834 | **33.101 s** ± 24.391 |

Table 3: **Stage-wise wall-clock time (s; mean ± std).** N/A indicates that the stage does not apply to the method. While VOYAGER is faster in several individual stages, it often incurs additional LLM refinement loops; CEDAR performs one-shot DFA construction and verification and then reuses skills symbolically.

## A.2 WHEN DO REGULAR LANGUAGES SUFFICE FOR SKILLS?

Let $\Sigma$ be the task-specific sub-alphabet extracted from logs by fixed monitors (e.g., `has_k(item)`, `time=night`, `in_biome(x)`). A skill is a policy that induces a set of feasible event sequences $L \subseteq \Sigma^*$.

**Assumption 1 (Observable milestones).** Numeric conditions are exposed via *thresholded* monitors (e.g., $has\_ \geq 3\_cobblestone$) and resource predicates change finitely often during a skill.

**Assumption 2 (Bounded subtask horizon).** Each subtask either completes or fails within $H < \infty$ event steps, after which control switches to a new subtask (possibly via a new alphabet).

**Proposition 1 (Regularity under milestone abstraction).** Under Assumptions 1–2, the set of successful traces for a single skill is regular, i.e., there exists a DFA $\mathcal{A}$ over $\Sigma$ such that $\mathcal{L}(\mathcal{A}) = L$. *Sketch.* With thresholded monitors, the event alphabet is finite; bounded horizon prevents unbounded counting. The induced control graph over milestone states is finite; accepted traces correspond to paths to accepting nodes.

**Proposition 2 (Composition closure).** If $L_1$ and $L_2$ are regular languages over compatible alphabets, then conjunction (intersection) and sequential composition (concatenation with $\varepsilon$-merging) yield regular languages. Hence specification enforcement by intersection and skill chaining by concatenation preserve regularity.

## A.3 LIMITATIONS

Our approach introduces several assumptions and limitations that warrant discussion:

**Ambiguity in natural language.** While our method does not assume human specifications are perfectly accurate, it relies on the ability of humans to provide correct counterexamples when the learned DFA misaligns with their intent. This assumes that humans can consistently judge whether a sequence matches their intended specification, which may not hold in cases of subtle or ambiguous semantics.

**Residual LLM hallucinations.** Although the LAPR algorithm can handle noisy membership queries and both the environment and verifier can provide counterexamples, our method cannot fully eliminate LLM hallucinations. If both the human and LLM share a similar misunderstanding of a task, the resulting specification DFA may be incorrect. Thus, while hallucination effects are mitigated, they are not completely resolved.

**Limited evaluation iterations.** Our experimental results are based on five runs per baseline to evaluate performance in Minecraft. While this is generally sufficient in the Minecraft setting—where each generated world presents substantial complexity for tasks like diamond mining—it introduces some variability in results. Due to the high cost of querying OpenAI APIs, we were unable to run more extensive trials.

## A.4 RAG

### A.4.1 RAG IMPLEMENTATION

Our RAG system is designed to enhance the reasoning and generation capabilities of language models by integrating structured knowledge retrieval. It leverages a database of pre-processed text chunks or symbol descriptions, embedding them into a vector space for efficient retrieval. The system supports multiple retrieval methods, including k-Nearest Neighbors (kNN) and Elasticsearch-based indexing, allowing for flexibility based on the deployment environment and use case.

The pipeline begins by chunking input data into manageable pieces, ensuring compatibility with the model's token limits. Each chunk is embedded using a state-of-the-art embedding model, capturing semantic relationships for downstream retrieval. These embeddings are stored in a database alongside their corresponding chunks. For retrieval, the system compares the embeddings of the user query against the stored embeddings, either through kNN for cosine similarity or via Elasticsearch's text search capabilities. This ensures highly relevant results tailored to the query context.

The system also ensures robustness by incorporating mechanisms to rebuild and maintain consistency between embeddings and the database. For instance, when new data is added or existing data is modified, the embeddings and retrieval models are updated to reflect the changes accurately. Additionally, the system includes mechanisms to index data into Elasticsearch for faster retrieval in scenarios involving large datasets.

To handle symbol-specific tasks, a specialized module allows for the addition and retrieval of symbols, including their semantic descriptions. Symbols can be retrieved based on their similarity to a query or used in downstream tasks to generate context-aware responses.

Finally, the system integrates with language models for generating augmented responses. By appending relevant retrieved chunks or symbols as context to the input query, it ensures that the language model produces more accurate and knowledge-grounded outputs. This approach makes the system suitable for tasks that require precise reasoning, such as answering domain-specific questions or solving complex problems. The use of both structured and unstructured data ensures flexibility and adaptability across a wide range of applications.

### A.4.2 RAG PROMPTS

This is a prompting example we used in our RAG system.

```
{"role": "user", "content": "For this sub-goal (specification): \"Mine[
    Log]: Mine a wood log from a nearby tree in the jungle biome.\", what
     is the most appropriate object? You are currently located at
    position (x: 4.50, y: 90.00, z: 25.50) in a jungle biome. It is
    facing yaw: -3.14 and pitch: -1.57. You have health: 20, food: 20,
    and saturation: 5. The current time of day is day. Your velocity is (
    x: 0.00, y: -0.08, z: 0.00). Nearby entities include: a parrot at
    19.77 blocks away, a chicken at 23.00 blocks away. You are surrounded
     by blocks such as stone, dirt, grass_block, coal_ore. Since the last
     observation, you have lost 1 of dirt."}
```

### A.4.3 RAG PERFORMANCE ANALYSIS

To evaluate the effectiveness of our RAG system in constructing a correct alphabet, we conducted a series of tests. The RAG system is provided with a task description (specification) and tasked with retrieving relevant symbols from the global alphabet. For the ground truth alphabet, we use the alphabet derived from skill DFAs that have been validated in the Minecraft environment, ensuring the correctness of the labels.

To compare the retrieved alphabet with the target alphabet, we use two metrics. The first metric is Absolute Accuracy, which measures the proportion of symbols in the target alphabet $\mathcal{A}^t$ that are correctly predicted in the retrieved alphabet $\hat{\mathcal{A}}$. It is defined as:

$$\frac{|\mathcal{A}^t \cap \hat{\mathcal{A}}|}{|\mathcal{A}^t|}$$

The second metric is the Overlap Coefficient, which calculates the size of the intersection divided by the size of the smaller set:

$$\frac{|\mathcal{A}^t \cap \hat{\mathcal{A}}|}{\min(|\mathcal{A}^t|, |\hat{\mathcal{A}}|)}$$

We evaluated our RAG system on a subset of 44 skill DFAs. The system achieved an Absolute Accuracy of $0.9372$ and an Overlap Coefficient of $0.9208$, both with a standard error of $0.10$. These results indicate that the retrieved symbols are highly similar to the target alphabet, providing a strong guarantee for the RAG system to construct a correct alphabet for task specifications.

To further assess the effectiveness of the text embeddings used in the RAG system, we compared the calculated text embedding similarities $D$ with the predicted results $X_i \leftarrow \hat{\mathcal{A}}_i \in \mathcal{A}^t$ using cosine similarity:

$$\frac{X \cdot D}{||X||||D||}$$

The RAG system achieved a cosine similarity score of $0.45$ (range $[-1, 1]$) with a standard error of $0.14$, demonstrating that the retrieved results are highly relevant to the query task.

| Metric | Absolute Accuracy | Overlap Coefficient | Cosine Similarity |
|---|---|---|---|
| RAG System | $0.9372 \pm 0.10$ | $0.9208 \pm 0.10$ | $0.4500 \pm 0.14$ |

Table 4: **RAG Alphabet Construction Performance:** The results are presented as average $\pm$ standard error.

### A.5  HUMAN GIVEN COUNTER-EXAMPLES

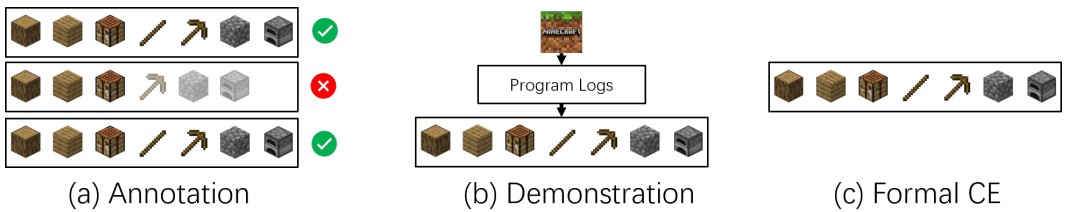

(a) Annotation      (b) Demonstration      (c) Formal CE

Figure 9: Three Ways for Human to Give Counter-Examples

Humans can provide counterexamples (CEs) in 3 ways:

1. **Annotations:** Humans can review videos or trajectories of the skills practiced by the CEDAR agent in the real environment and mark incorrect trajectories. These marked trajectories are then used as CEs.

2. **Demonstrations:** Humans can provide demonstrations by playing Minecraft. The human actions are recorded in the program logs, which can be converted into formal CEs.

3. **Formal Counterexamples:** For simpler DFAs that can be visualized as graphs, humans can directly provide formal CEs by inspecting these graphs.

### A.6  SIMULATION COUNTER-EXAMPLES

To further evaluate the correctness of the learned skills and their alignment with human specifications, we simulate these skill DFAs in the real environment and refine them using counter-examples collected during the process. However, due to the complexity of the environment, some corner cases may not be encountered by the agent within a limited number of iterations. To address this, we conducted experiments to measure the success rate of collecting counter-examples.

For the experimental setup, we first generated incorrect DFAs by randomly adding or removing transitions from correct skill DFAs. The skill DFAs selected for this experiment are designed to

| Item | Accuracy | Standard Error |
|------|----------|----------------|
| Dirt | 0.9727 | 0.1629 |
| Birch Log | 0.8636 | 0.3432 |
| Grass Block | 1.0000 | 0.0000 |
| Birch Leaves | 0.9909 | 0.0949 |
| Stone | 0.9727 | 0.1629 |
| Coal Ore | 1.0000 | 0.0000 |
| Iron Ore | 1.0000 | 0.0000 |
| Copper Ore | 0.9909 | 0.0949 |
| Gold Ore | 0.9636 | 0.1872 |
| Redstone Ore | 0.9636 | 0.1872 |
| Emerald Ore | 0.4909 | 0.4999 |
| Diamond Ore | 0.9818 | 0.1336 |
| Lapis Ore | 0.9636 | 0.1872 |
| Andesite | 0.9818 | 0.1336 |
| Granite | 0.9636 | 0.1872 |
| Sand | 0.8727 | 0.3333 |
| **Average** | 0.9358 | 0.1692 |

Table 5: **Success Rate and Standard Errors of Counterexample Discovery in Minecraft Simulations.** The table shows the accuracy and standard errors for different items.

locate specific objects and collect them, providing a practical context for evaluating the success rate of counter-example discovery. Since these modified DFAs do not match the dynamics of the real environment, counter-examples must exist. We then simulated these DFAs in the environment to identify whether any counter-examples could be collected. For each DFA, we simulate it 110 times. A counter-example occurs when the DFA's behavior diverges from the expected outcome in the real environment. For instance, consider the `mine_stone` DFA, which is expected to collect a cobblestone upon reaching its accepting state. If, during simulation, the accepting state is reached but no cobblestone is present in the bot's inventory, this trajectory constitutes a counter-example.

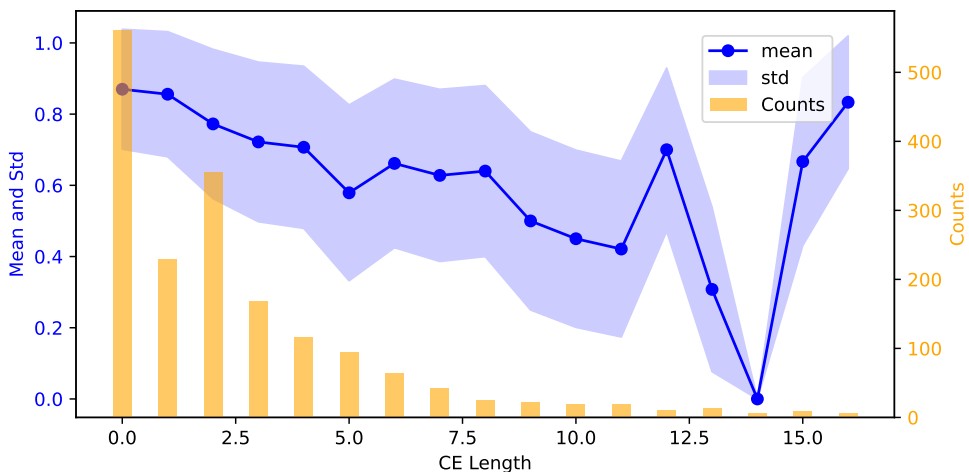

Figure 10: Mean, Std of CE Collection Probability with Lengths of CEs

The results in Table 5 demonstrate that the RAG system effectively identifies counterexamples during DFA simulations in Minecraft, with most items achieving an accuracy higher that 0.96 and a standard error less than 0.2, indicating consistent detection. Notable exceptions include Birch Log and Sand, which achieved an accuracy higher than 0.86 with a standard error around 0.3, and Emerald Ore, which had the lowest accuracy at 0.49 with a standard error of 0.4999. These variations

highlight the challenges of certain items in aligning with the DFA dynamics. On average, the system achieved an accuracy of $0.9358$ with a standard error of $0.1692$, underscoring its overall reliability and precision in identifying counterexamples across diverse scenarios.

We observed in Figure 10 that the probability of collecting CEs decreases as the length of the CEs increases. This is because shorter CEs indicate that the skill DFA fails early in its execution, requiring fewer interactions with the environment. In contrast, longer CEs suggest that the skill DFA is mostly correct, with errors occurring only after extended interactions with the environment. However, this is not a significant concern, as the majority of CEs are short, with lengths less than 7. Within this range, the probability of collecting a CE is consistently above $0.6$, ensuring that CEs can reliably be collected within multiple simulation attempts.

## A.7 ITHOR LLM QUERY COST

Each iTHOR skill required exactly **two** LLM calls: (i) one to construct the task-specific sub-alphabet (via RAG) and (ii) one to synthesize the code-based oracle program that binds monitors and action stubs. No human-provided counterexamples were used. All tasks succeeded in a single pass *except* COOK_POTATO, which initially failed due to an unfavorable camera perspective; the EQ-oracle surfaced a counterexample and we resolved it by inserting a single LOOKUP action into the DFA—this fix *did not* require additional LLM queries.

We do not report VOYAGER on iTHOR because the released VOYAGER implementation is tightly coupled to Minecraft/Mineflayer APIs and does not provide an iTHOR-compatible action interface, making a direct, controlled comparison infeasible.

| Task | LLM Queries | Initial Outcome | After CE Fix |
|---|---|---|---|
| MAKE_1_TOAST | 2 | success | — |
| MAKE_COFFEE | 2 | success | — |
| COOK_POTATO | 2 | fail | success (no extra LLM) |
| MAKE_SALAD | 2 | success | — |
| STORE_PLATE_IN_FRIDGE | 2 | success | — |

Table 6: **iTHOR LLM query budget.** Two calls per skill: one for sub-alphabet construction and one for code-based oracle program generation.

