# OpenReview forum: "CEDAR: A Counter-Example Driven Agent with Regular Restriction"
_ICLR.cc/2026/Conference — ICLR 2026 Conference Withdrawn Submission_

### Official Review · Reviewer_zoyR · 2025-10-26

**Soundness:** 3
**Presentation:** 2
**Contribution:** 2
**Rating:** 4
**Confidence:** 3

**Summary:**

This paper introduces the CEDAR framework, which integrates active automata learning with reinforcement learning to enforce formal specifications during policy learning. When agent behavior violates a predefined specification, CEDAR utilizes the resulting trajectory as a counter example. Employing an L* like query based algorithm, it learns a finite automaton, termed a regular restriction, that formally captures the violating behavior pattern. This learned automaton subsequently constrains the agent's policy space, precluding the recurrence of behaviors recognized by the restriction. Through iterative refinement involving counter example generation, restriction learning, and policy constraint, CEDAR facilitates the learning of policies that are demonstrably compliant with the specification while optimizing for the task objective.

**Strengths:**

1. Integrating active automate learning with RL for handling safely specifications is a promising direction. The idea is intuitive.
2. Unlike many LLM agents focused solely on goal completion, CEDAR treats adherence to human specifications as a core objective, providing mechanisms (Verifier, DFA intersection) for enforcement.
3. Experiments show this method is effective in following human specifications.

**Weaknesses:**

1. This is minor, line 112, NL2LTL, NL->LTL, please be consistent.
2. Figure 1 is good, but what are CE, EQ, MQ? Please exmaplain these concepts breifly in the catption. Readers may not have clear understanding when they read Figure 1 at the first time.
3. Scability of CEDAR is concerning but not analyzed in the paper. Please demonstrate how efficient are the DFA learner and Verifier, and what are the limitations and potential failing cases.
4. I understand this paper contributes to various domains, but I would like to see a clear description of the problem formulation (including RL, safety RL, Finite Automaton, etc). I believe this helps readers understand how your different framework components and algorithmatic advancements contribute to the whole problem. The writing is not concentrated currently.
5. Many fileds in related work are missing. Given the topic of this paper, it would be better to include related work from safty RL, counter-example learning, etc.
6. This is minor, please cite the models and embeddings you used in the paper.

**Questions:**

1. The effectiveness of the framework is heavily rely on DFA learner. Do you have analysis on the correctness of LLM that answers membership queries (MQ)? What are the failing mode?
2. The limitation of this work is discussed in Appendix, but I would like to see some *examples* of these limiations. This could help the entire community.

---

### Official Review · Reviewer_cUY9 · 2025-10-30

**Soundness:** 2
**Presentation:** 2
**Contribution:** 4
**Rating:** 4
**Confidence:** 3

**Summary:**

This paper proposes a novel method for sequential decision making in an open world environment, when given textual description of the objective to achieve and constraints and guidance on how to achieve it. The key novelty of the proposed method, called CEDAR, is that it attempts to learn skills that are represented as DFAs. This works by first using a RAG to select the alphabet needed for achieving the given objective, then using a DFA learning algorithm using an LLM to perform membership queries (i.e., is the action consistent with the guidance given by the human) and interactions with the environment to perform equivalence queries (i.e., can I perform the planned action in the environment and a counter example if you cannot).
The authors compared CEDAR with VOYAGER on Minecraft task and (minimally) on iThor tasks, showing improved performance and better adherence to the human’s instructions.

**Strengths:**

The idea is novel and exciting. I think this has a lot of promise, and the evaluation is also reasonable.
The concept of learning DFAs as an anchor for alignment and using DFA learning algorithm is very interesting. Similar to LTL-based solutions, but the authors clarify well the differences.
The comparison to VOYAGER is also adequate and shows good results.

**Weaknesses:**

While I'm excited about the general research direction, I believe the paper requires significant re-writing to be clear and self-contained. At this point, it feels to me more like a technical report or Arxiv paper than a well-written and of the academic writing level expected from a high-quality conference paper.

I list below examples:

Line 78: “… using a human-in-the-loop learning paradigm ….” – I am confused. Is there a human in the loop or is it that you use the LLM to estimate what the human would say based on the instructions given by the human?

Line 80: “The output formal specification are given to the DFA learner.” – which output forma specification? How is it given to the DFA learning? This is vague and not clear (even after reading the paper).

Line 87: “(i) a minimal canonical structure with closure under intersection …” – what do you mean by “canonical”, “minimal”, and “closure under intersection” in this context? The authors show explain more
clearly what they mean.

Lines 91-96:  These lines are not coherent to me, and look more like a set of keywords then actual sentences. What do you mean by “regular-language scaffold?” what is an “embodied skill” here? Or that the LLMs “ground intent”?

Line 101: What is an “minimally adequate” teacher?

Line 120: what do you mean by “canonical temporal controllers”?

Line 147: What is are “event-monitor symbols”?

Line 154: “DFAs provide minimal, ….” – minimal in what sense?

Line 160: What is a “specification-decomposition step”?

Line 201: What is a “unified active” loop?

See below my questions, which are also pointing to where the paper is not clear enough.

As a minor issue, the citation format is incorrect in most of the paper, e.g., “Wu et al. (2022);” instead of (Wu et al., 2022). This is easily fixed by using the correct citation command.

**Questions:**

1.	In line 149 you mention a RAG being used to return a finite subset of the global alphabet. How has this RAG been implemented? Was it tuned to the task at hand in some way? How does it know which alphabet elements to return to a given task?

2.	In line 186 you write that “We analyze the erroneous DFA and reuse EQ counterexamples”. How is this done?

3.	In line 212: how can a counterexample indicate that a symbol is missing? Is it when a state includes a symbol that is not in the set the RAG returned?

4.	Do you assume that the environment is deterministic? That is, if a skill has successfully worked, does it mean it must also work in the next time it is applied?

5.	Can you formally define what you mean by a “verb” here? Is this an action you can tell the agent to perform?

6.	In line 248: how can we detect a “missing action”? which logs are you referring to? Aren’t you executing the “policy” directly on the environment?

7.	In the Verifier section, the authors write that when a violation between the skill and the specification is detected, then the alphabets are merged and the intersection of the skill and the specification are taken. How is this done exactly? You later explain formally how to merge skills but not how to “merge” skills and specifications.

8.	I like a lot the fact that you can chain skills but I am not clear on who initiates this. Is this done by the LLM? Or is this done by CEDAR when something happens? How do you decompose the specification and goal given by the user to the list of skills you want to chain?

---

### Official Review · Reviewer_q4sV · 2025-10-31

**Soundness:** 3
**Presentation:** 3
**Contribution:** 2
**Rating:** 4
**Confidence:** 3

**Summary:**

This paper introduces a novel framework that combines LLM with formal methods, specifically using DFA to represent both skills and human constraints. This paper proposes to encode skills as regular languages using DFA, which is rigorous for agent behavior that enables verification and composition. The proposed framework uses RAG to create task-specific sub-alphabets from large action spaces, which makes learning tractable in complex environment.Through experiments, the proposed framework outperforms Voyager in terms of constraint adherence, task completion, and efficiency.

**Strengths:**

1. The DFA based approach provides formal guarantees about agent behaviors, which also support constraint enforcement and skill chaining.
2. The counterexample loop allows continuous refinement and alignment with human specifications.

**Weaknesses:**

1. CEDAR requires more upfront LLM interactions to learn new skills, though this is amortized through reuse.
2.  Experiments used only five trials per baseline due to API costs, introducing some result variability.

**Questions:**

The evaluation is limited to small scale. I would like to see more analysis and experiment result.

---

### Author Response · Authors · 2025-12-02
**General Response**

We thank all reviewers for their thoughtful and constructive feedback. We appreciate that the core idea—representing both skills and human specifications as DFAs learned through an LLM-assisted, counterexample-driven loop—is viewed as novel, promising, and relevant for neurosymbolic AI and alignment. Below we address the major concerns raised across reviews and outline how we will revise the paper. We focus only on the substantive points, as requested.

---

## **Clarity, Self-Containment, and Organization**

A recurring concern is that the paper currently assumes too much background and introduces many technical notions too abruptly (e.g., MQ/EQ/CE, minimally adequate teacher, regular restrictions, event-monitor symbols, decomposition of specifications). This makes parts of the method difficult to parse.

In the revision, we will:
- Add a concise **Problem Formulation** section early in the paper that clearly explains:
  - the RL setting (Minecraft/iTHOR),
  - how event logs induce words,
  - how DFAs serve as formal representations of skills and specifications, and
  - how the active learning loop (MQs from the LLM, EQs from environment/human) fits together.
- Introduce a **glossary** of central concepts (alphabet, verb, object, event-monitor symbol, regular restriction, skill DFA vs. specification DFA).
- Rewrite the main pipeline description so that readers can understand Figure 1 in a top-down manner: what the LLM does, what the environment/human does, and how the DFA learner, Skill Manager, and Verifier interact.

This restructuring resolves most of the line-level confusions mentioned by reviewers.

---

## **Positioning and Novelty**

The reviews also ask for clearer differentiation from related paradigms such as:
- code-generating agents (Voyager),
- NL→LTL methods,
- safe RL and shield-based approaches.

We will strengthen the discussion to highlight that:
- CEDAR deliberately uses **minimal DFAs over a task-specific alphabet**, enabling exact intersection, composition, and reuse that code-level agents do not provide.
- Unlike NL→LTL pipelines, CEDAR’s specifications and skills are **learned on demand** through counterexamples rather than fixed ahead of time, making repair and refinement natural.
- Unlike shielded RL, CEDAR’s constraints are not hand-written; they are **actively learned regular restrictions**, with LLM-driven MQs and environment-driven EQs.

We will expand the related work section to cover missing areas (safe RL, counterexample-guided RL, automata learning), and articulate the conceptual differences more explicitly.

---

## **Scalability and LLM Query Cost**

A major concern is whether CEDAR is scalable given that learning DFAs requires LLM interaction.

Our existing results already show:
- **LLM cost is front-loaded and amortized**: once a skill DFA is learned, it can be reused, intersected with specifications, or concatenated with other skills without further prompting.
- **CEDAR uses fewer total LLM calls and less wall-clock time** than Voyager on long-horizon tasks once skills are cached.

These results currently appear in the appendix; in the revision, we will move the key tables/plots to the main text and frame them as a direct answer to the scalability concern. We will also more clearly describe how RAG-based alphabet restriction keeps the DFA learning problem tractable.

---

## **Correctness and Failure Modes of the LLM MQ-Oracle**

Another central point concerns whether the LLM can reliably answer MQs.

We follow a **probabilistic MAT** viewpoint: the LLM may be wrong, but errors are corrected through environment-supplied counterexamples and consistency checks in LAPR. To make this explicit, we will:
- Summarize in the main paper the RAG-alphabet evaluation, which shows that the LLM reasons over a meaningful subset of the global alphabet.
- Move into the main text the counterexample-discovery experiments, which show that environment interactions reliably expose erroneous DFAs.
- Add concrete examples of failure modes (e.g., ambiguous NL specifications, long-horizon rare events, systematic misunderstandings shared by the human and LLM).

This provides a clearer justification for why the combined LLM+environment teacher is practically workable.

---

## **Limits and Examples of When CEDAR Fails**

Reviewers requested more concrete limitations. We will expand the Limitations section with explicit cases.

---

## **Evaluation Scale**

All reviewers noted that the empirical section is limited in scale (few tasks, five trials per method). We acknowledge this limitation. For the next revision, **we will add more experiments, larger-scale evaluations, more repeated runs, and additional ablations**, and then resubmit with a significantly strengthened evaluation section.

---

We will revise the paper accordingly and resubmit with clarified structure, better-motivated design choices, and expanded experiments.

---

### Note · Authors · 2025-12-02

I have read and agree with the venue's withdrawal policy on behalf of myself and my co-authors.